# Beliefs, perceptions, and behaviors impacting healthcare utilization of Syrian refugee children

Riham M. Alwan[1,2,3]*, Daniel J. Schumacher[1,2], Sevsem Cicek-Okay[4], Sarah Jernigan[5], Ahmed Beydoun[1], Tasnim Salem[6], Lisa M. Vaughn[1,2,5]

1 College of Medicine, University of Cincinnati, Cincinnati, OH, United States of America, 2 Cincinnati Children's Hospital Medical Center, Cincinnati, OH, United States of America, 3 University of California San Francisco, San Francisco, CA, United States of America, 4 College of Arts and Sciences, University of Cincinnati, Cincinnati, OH, United States of America, 5 College of Education, Criminal Justice and Human Services, University of Cincinnati, Cincinnati, OH, United States of America, 6 Independent Consultant, Dallas, TX, United States of America

* riham.alwan@ucsf.edu

**Data Availability Statement:** All relevant data are within the manuscript and its Supporting Information files.

## Abstract

### Background

Approximately 18,000 Syrian refugees have resettled to the United States. Half of these refugees are children, whose age and refugee status jeopardize their abilities to attain quality healthcare. Information on Syrian refugees' health in the U.S. is limited. This qualitative study sought to explore Syrian refugee parents' beliefs, perspectives, and practices regarding their children's health through in-depth interviews.

### Methods

Eighteen Syrian refugee parents residing in Cincinnati, Ohio were interviewed in Arabic by bilingual researchers using semi-structured in-depth interviews. The interviews were recorded, transcribed, and translated. Three members of the research team independently coded each interview using an inductive thematic analysis approach.

### Results

Analysis identified four salient themes: stressors preclude health seeking behaviors, parents perceive health barriers, parents are dissatisfied with the healthcare system, and parents use resilience behaviors to overcome barriers. Stressors included poor housing and neighborhoods, reliving traumatic experiences, depression and anxiety, and social isolation. Dissatisfaction included emergency room wait times, lack of testing and prescriptions. Health barriers included missed appointments and inadequate transportation, translation services, health literacy and care coordination. Parents reported resilience through faith, by seeking knowledge, use of natural remedies, and utilizing community resources.

**Funding:** The authors received no specific funding for this work.

**Competing interests:** The authors have declared that no competing interests exist.

**Abbreviations:** ED, Emergency department; PTSD, Post traumatic stress syndrome.

## Conclusion

This qualitative study provides information on the beliefs, practices, and behaviors of Syrian refugee parents related to health care utilization of pediatric refugees in the United States. Psychosocial and environmental stressors as well as perceived systemic health barriers, hinder health seeking behaviors in Syrian refugee parents. Culturally relevant care targeting perceived barriers and incorporating resilience behaviors may improve parental satisfaction and parental health seeking behaviors. Further study is needed to implement and evaluate interventions that target identified barriers.

## Introduction

There are an estimated 5.6 million Syrian refugees worldwide, with 18,000 Syrian refugees resettled to the United States of America since the Syrian civil war began nine years ago [1]. Refugees relocated to non-traditional migration destinations, such as a Cincinnati, experience significant challenges in acclimation and adaption [2, 3]. Arriving with various skillsets and levels of literacy, refugees have unique health needs, often tied to their transit experiences and countries of origin. As a result, refugees often struggle with unmanaged acute and chronic conditions, including mental health diseases [4].

Factors influencing refugee health are complex and complicated by transit experiences, prior trauma, poor living situations and barriers to healthcare access [4–7]. Health barriers identified by refugees in Europe include time pressure, linguistic and cultural differences, and lack of continuity of care [5]. Refugees' experiences with primary care physicians reveal they feel stigmatized with linguistic barriers and find difficulty navigating the system, which complicates health seeking behaviors [8, 9]. Middle Eastern asylum seekers commonly suffer from somatic problems, most often neurologic and gastrointestinal [10–12]. Mental health issues among Syrian refugees are common given their traumatic experiences of escaping a dictatorial and economically depressed regime [7, 13–16]. Refugees with high rates of post-traumatic stress disorder and depression have been described as having low levels of help-seeking behavior [17].

While some understanding of the health needs and behaviors of Syrian refugees in other countries exist, little is known about Syrian refugees in the U.S. Over half the Syrian refugees in the U.S. are children, whose age and refugee status jeopardize their abilities to attain quality healthcare. Refugee resettlement agencies provide 3 months of services, including a mandatory medical screen exam within 30 days of arrival to the U.S. Establishment of a medical home is not mandatory and varies throughout the U.S. Challenges to health care access and navigation are likely exacerbated in non-traditional migration cities given the lack of immigrant support infrastructure [3, 18, 19]. Non-traditional migration cities have been shown to be associated with poorer health outcomes among minorities [2, 20]. This qualitative study sought to explore the parental beliefs, perspectives, and practices impacting health and healthcare utilization of Syrian refugee children in a non-traditional migration destination. Understanding in this area will improve healthcare providers' ability to provide quality pediatric care that meets the needs of this growing and vulnerable population.

## Materials and methods

The Institutional review board of Cincinnati Children's Hospital Medical Center approved of this study as a non-human subjects determination. Each participant was consented in written form in Arabic and English.

## Setting

Cincinnati is one of many refugee resettlement cities across the U.S. Previous research has described Cincinnati as a "non-traditional city of migration," defined as a city with less than 2.5% growth rate from 2000 to 2010 of a particular ethnic group [2, 3, 20–23]. Non-traditional migration cities lack immigrant infrastructure that facilitates immigrants' basic and social needs. There is a paucity of community centers, faith-based organizations, ethnic grocery stores, intricate public transportation, and social support networks in such cities. Additionally, these cities do not have robust social services, such as legal aid, financial assistance, and unemployment services, that serve immigrants and refugees [20, 21]. Poorer health outcomes have been associated with minorities living in non-traditional migration cities [2, 20].

## Participants

At the time of study, Syrian refugees in the Greater Cincinnati area included 45 adults and 46 children under age 18, with a total of 19 families. All participants were recruited via phone. Ten of the 19 families met inclusion criteria. Nine of the 10 eligible families were enrolled, with a total of 18 participants. Inclusion criteria were: participants of Syrian origin; having dwelled in Syria prior to the civil war; cohabitating with children or grandchildren below the age of 18; refugee status; Medicaid enrolled or eligible; and migration within the last two years. Participants who did not speak Arabic or English were excluded. Only primary care givers were interviewed in the study, as no other elderly house-hold members were present. All participants were initially resettled by the same resettlement agency in Cincinnati to the same neighborhood. The first few participants were initially recruited through a social service agency that serves Syrian refugees. Snowball recruitment, where existing subjects recruit other participants, was utilized to increase number of participants [24]. Participants received $50 gift card incentive for their participation.

## Data collection

Semi-structured interviews, ranging from 45–90 minutes, were conducted in Arabic by bilingual investigators (RA and AB) between April and Sept 2017 (S1 File). The open-ended interview guide was developed and vetted by field experts, local direct providers, and other Syrian refugees (see Appendix A). The guide was pilot tested with 4 local Syrian refugees ineligible for this study. Interviews explored the beliefs, practices, and expectations regarding health seeking among Syrian Refugee parents in the emergent and non-emergent clinical settings. The researcher's statement of positionality is available for review (S2 File). All study procedures were determined to be exempt by the Cincinnati Children's Hospital Medical Center Institutional Review Board. Ethical standards were upheld through the use of informed consent, confidentiality, and ability to withdraw at any point.

## Data analysis

Interviews were audio-recorded in Arabic initially. Interviews were simultaneously translated and transcribed into English and transcribed by a licensed interpreter. Two bilingual researchers (RA, AB) double checked the translations for accuracy and content. Inductive thematic analysis is the qualitative methodology that was used to analyze the transcripts. A code is a label assigned to concepts found in the interviewee's narrative. Three members of the research team (RA, SJ, SCO) independently reviewed and coded each interview upon completion of data collection, using the support of Dedoose software [25]. A codebook was then built with agreement of the research team. Subsequently, investigators independently placed codes into

broad categories. Through an iterative process and a series of bimonthly meetings, the group reached consensus on the categories and continued to independently code the interviews. Finally, the group convened for a collective compilation of categories into themes [25, 26]. Participant enrollment continued until saturation of themes occurred and no new information came forth, which is a standard procedure for estimating sample size for qualitative studies [25, 27]. A thorough process of member checking, which involves reviewing the data and the results with participants to check for accuracy and resonance with their experiences, was performed to ensure the credibility and trustworthiness of the findings. Measures were taken to promote credibility, transferability, and confirmability [28–30]. The emergent themes of this study were reviewed by local community stakeholders, national experts, and the study participants.

## Results

### Demographics

Ninety percent of eligible Syrian refugee families were enrolled. Nine parent dyads, nine males and nine females, were interviewed. Parental age ranged from 22 to 46 years. Children's ages ranged from 4 mo to 19 years, with a range of 2–7 children per family. Two pregnant mothers were interviewed. Educational level of participants varied from 0 to 9 years of formal schooling. Unemployment was high, with only 1 female and 6 males employed full-time.

At the time of interview, all refugee parents had been living in the U.S. for less than 2 years, with a range of 3–14 months, average of 7.8 months. All participants originated from different regions of Syria, including both urban and rural settings, and immigrated to another Arabic-speaking country prior to their placement to the U.S. Six out of 18 participants spent time on a refugee camp. Four participants spoke another language in addition to Arabic: Kurdish and Turkish. None of the participants were fluent in English.

Four salient themes were identified. Quotations illustrating each theme and subtheme are found in corresponding tables. Findings did not differ among educational level, employment status, or Syrian area of origin.

### Theme 1: Stressors preclude health seeking behaviors

Refugees reported experiencing stressors that precluded them from seeking care and maintaining their children's health. Stressors were both environmental and psychosocial, as seen in Table 1.

### Poor housing and neighborhoods

Participants reported that one of the most significant barriers to their health seeking behaviors are their poor housing and neighborhoods. Due to unsanitary and unsafe conditions, parents felt current housing was uninhabitable for their children, particularly those with preexisting medical conditions. They described hearing gunshots and witnessing drug activity in their neighborhoods which limited their mobility. Hearing and seeing wild animals around and in their houses (e.g. snakes, raccoons) upset them. Syrian refugees feared leaving their homes and did not seek help for medical conditions. Parents' concern with poor housing and neighborhoods distracted them from focusing on their children's health. One mother shared, "We did not dare to go out. Even when we needed to go out to get food, the kids would be too afraid to leave the house." With the assistance of volunteer agencies and community members, every family had relocated from its original assigned housing at the time of interview, expressing various levels of satisfaction with current housing and neighborhoods.

**Table 1. Stressors preclude health seeking behaviors.**

| | |
|---|---|
| **Poor Housing and Neighborhoods** | • *It was like you were entering a sheep pen, not like you're entering an apartment. It smelled like ass, like ass. (3B)* |
| | • *They put me in a place that was a bad neighborhood. In terms of the neighbors, there were problems. They come and knock on the door wanting food, wanting something to drink. They're always outside getting high and whatnot. The place was completely full of bugs, and it was a dirty place. (3A)* |
| | • *The home they placed us in was not good at all. It affected our emotional state. There was fear. . .We took them out of a frightening place and brought them to another place that gave them no sense of security. We left Syria due to the fear, and killings, and oppression. We went to Lebanon and still did not feel safe. We then came here for the sense of security but still did not find it. Our house was horrendous. The neighborhood was scary. We felt we couldn't go out much for that reason. We felt like we were locked up and restrained at home. (4B)* |
| | • *When we first arrived, we were placed in a really bad home. It was filled with snakes, bugs, creatures, and raccoons. . .the children all became depressed. My daughter got sick for that reason. . . We already had to deal with the war and now there was this added stress? (5A)* |
| | • *We did not feel safe and secure here at all. When we first arrived we had shots fired on our home. We had to keep the windows and doors locked constantly. We did not dare to go out. Even when we needed to go out to get food, the kids would be too afraid to leave the house. (5B)* |
| **Reliving Traumatic Experiences** | • *We have seen so much. A person imagines everything that has happened, questions . . ..Nothing is going to change the past. . .I start crying. I start to remember. I cannot focus on the lesson, on anything. (7B)* |
| | • *It is worse than a prisoner because a prisoner knows that he is locked up. However, to have your door open and the ability to be free but to feel like a prisoner and live in fear. (2A)* |
| | • *When I am not feeling well. . .mentally and emotionally. . .I am not in the mood to learn. I don't have it in me to decide this or that. (8B)* |
| **Depression and Anxiety** | • *The [children] reached a point where they became mentally exhausted. . .All the stress they have inside will only cause them to become ill. Emotional stress is much worse than physical stress. Emotional stress on the inside can really destroy a person. (4B)* |
| | • *"A refugee is not in the same psychological state as a native. . . This refugee will have mental illnesses due to the war and having to migrate." (2A)* |
| | • *These four months here have been unlike anything I have ever experienced before. The stress and burden it puts on a person. It's too much. How am I supposed to keep up with everything? (9B)* |
| | • *Life is really, really difficult here. This is the first time in my life I experienced this degree of suffering. From the doctors and the life here. The loneliness as well. Everything. Feelings of hopelessness and emotional suffering. It has all been a struggle. (5B)* |
| **Social Isolation** | • *I feel like I am blind and deaf in this country. . .I don't understand them. I can't talk to them. (9B)* |

## Reliving traumatic experiences

Previous traumatic experiences, being "refugees of war," affected parents' successful adaptation and acculturation. These traumatic experiences hindered health seeking behaviors, with the trauma negatively impacting their ability to learn a new language and new health system. Participants reported difficulty participating and concentrating in mandatory educational sessions offered by resettlement agencies and community agencies. Participants noted that their preoccupation with previous trauma affected their ability to remember medical advice and medical appointments for their children. A parent shared struggles with medical decisions, "When I am not feeling well. . .mentally and emotionally . . .I am not in the mood to learn. I don't have it in me to decide this or that."

### Parental depression and anxiety

Parental anxiety and depression hindered them from seeking help for their children's medical conditions. Parents described suicidal ideations and insomnia. They described feeling "pressure" that was distracting them from focusing on their children's health. One participant reported, "A refugee is not in the same psychological state as a native. . . This refugee will have mental illnesses due to the war and having to migrate." Parents shared feelings of task overload, affecting their ability to manage their children's health. "How am I supposed to keep up with everything," was a common phrase shared by multiple parents.

### Social isolation

Parents shared experiences of social isolation that exacerbated their inability to manage their children's health. They explained that lack of child care, transportation, and health literacy was exacerbated by a lack of social support network. Syrian refugee caretakers expressed that their limited English language skills preclude their ability to participate in social life in the U.S.

### Theme 2: Health barriers hinder parental health seeking behavior

Parents perceived health barriers hindering their ability to seek medical care and manage their children's preexisting medical conditions, as seen in Table 2.

Parents perceived health barriers hindering their ability to seek medical care and manage their children's preexisting medical conditions.

### Missed appointments

Parents reported missing medical appointments, including dental, vision, primary care, and subspecialty appointments. These missed appointments led to delays in diagnosis and treatment of their children's medical conditions. They explained that these missed appointments led to worsening children's health and feelings of inability to provide for their children. Refugees expressed challenges with scheduling appointments after they had been missed. Some parents attributed the missed appointments to difficulties with language, transportation, and navigating medical facilities. Language barriers and difficulty navigating the system caused this participant to miss his child's appointment, and he describes, "the phone calls are always automated. I keep trying to say, No English, Arabic. . . If I had an appointment, I would not know anything about it." Additionally, a second participant notes "When you miss an appointment like this, you need months till they can reschedule," highlighting the compounding problem of one missed visit.

### Inadequate transportation

Some refugee parents shared challenges with transportation to medical visits, both scheduled and emergent. One parent recollects that, "a refugee will not know how to go and where to go for the appointment." They reported difficulty navigating public transportation, as they did not know the routes or schedules of buses nor even how to obtain that information. One refugees reports, "I used to take the children [to doctor's appointments] walking back and forth. There is also a bus that used to pass by our house but we didn't know how to ride it and where to tell it to drop us off. . . One day we walked out of the hospital one day and it poured. What are we going to do? The kids walked in the cold. We basically took a shower in the street." Some refugees found greater ease meeting their health needs when they had acquired vehicles and driver's licenses.

**Table 2. Parents perceive health barriers.**

| | |
|---|---|
| **Missed Appointments** | • *The biggest problem is just the appointments. For example. I don't know how to make a doctor's appointment if I need to go. (8B)* |
| | • *We are missing appointments because of language difficulties. (1A)* |
| | • *I have been in this country for three months now. I missed many appointments and I do not understand anything. The phone calls are always automated. I keep trying to say, "No English, Arabic". I really struggled with this issue. . . If I had an appointment, I would not know anything about it. (5A)* |
| | • *When you miss an appointment like this, you need months till they can reschedule. (2A)* |
| **Inadequate Transportation** | *The transportation was difficult. (2B)* |
| | • *I used to take the children [to doctor's appointments] walking back and forth. There is also a bus that used to pass by our house but we didn't know how to ride it and where to tell it to drop us off. . .One day we walked out of the hospital one day and it poured. What are we going to do? The kids walked in the cold. We basically took a shower in the street. (9A)* |
| | • *A refugee will not know how to go and where to go for the appointment (2A).* |
| **Inadequate Translation Services** | • *Here we are really struggling with the translation. Even with an interpreter, we still find it hard to understand them at times. (5B)* |
| | • *The interpreter would be difficult to understand if it's over the phone . . . When an interpreter speaks on the phone–excuse my language- but it is despicable. . .The whole thing is despicable. The language. The attitude. Every single interpreter on the phone is extremely terrible (6A)* |
| | • *I was talking to the interpreter, and he fell asleep on the phone! I could hear him snoring! (4B)* |
| | • *My words were lost [6A]* |
| | • *If there is an interpreter everything will be solved [3A]* |
| **Lack of Health Literacy** | • *Every time I go they would give me more medicine. They don't even explain to me what the medication is for, what it helps with. Till now, I don't even understand anything from the doctors. (9B)* |
| | • *In terms of sick children, do I take them to the hospital or, no, just communicate with the doctor that follows their condition? (3A)* |
| | • *We do not know—we do not know—now, if a child gets sick, we do not know where we would take them. We do not know which medical center we would take him to. (1A)* |
| **Lack of Care Coordination** | • *There needs to be more communication between the hospitals. It should show in their system that this man has to take his son to his appointment on the 23rd at 4 pm. How am I supposed to make it to my appointment on the same day at 5 pm? (5A)* |
| | • *No one in the hospital is telling us how we can navigate the system. No one is helping us diagnose our problems. How to get treated for the illness we have. . . .A patient needs to understand their illness in order to get treated. (2A)* |
| | • *They're not coordinating with each other [3A]* |
| | • *They should have people assigned to take care of them [us]. They should have people assigned to each family that understands the situations they came from (5B)* |
| | • *They can make appointments for us, and if you don't know how to make an appointment, they [could] teach you how to make one. (4A)* |

## Inadequate interpreter services

Some refugee parents described challenges with both in person and video translation services provided in clinic and emergency settings. A participant stated "My words were lost," when discussing their experiences with interpreters. Additionally, due to differences in colloquial Arabic dialects, one mother noted: "Even with an interpreter, we still find it hard to understand them at times."

## Lack of health literacy

Health literacy is commonly defined as the degree to which an individual has the capacity to obtain, communicate, process, and understand basic health information and services, to ultimately make a healthcare decision. In this study, health literacy was not associated with education level. We found that refugee parents struggled to comprehend their children's diagnoses, treatment plans, and medication administration. Refugees discussed challenges with figuring out where to go and who to call for their children's various health needs. One woman notes, "now, if a child gets sick, we do not know where we would take them. We do not know which medical center we would take him to."

## Lack of care coordination

Refugee parents expressed that their children had various medical needs including primary and sub-specialty care. They reported challenges with coordinating appointments and sharing of information amongst the providers. One participant reflected that "no one is helping us navigate the system." Another suggested the need for care coordinators assigned to refugee families to assist in navigating the healthcare system. The parental need for care coordination was recurrent in all interviews preformed in this study.

## Theme 3: Dissatisfaction with the U.S. healthcare system is prevalent

Syrian refugee parents expressed appreciation for the care and compassion demonstrated by direct medical providers, such as nurses and doctors. However, they seemed dissatisfied with the health care system, as seen in Table 3.

**Table 3. Parents are dissatisfied with the healthcare system.**

| Emergency Department Wait Times | • *When I came to the emergency room and sat waiting for 5 or 6 hours, I got sick. I got emotionally and physically ill. Psychological illnesses increase the physical illnesses. (2A)* |
|---|---|
| | • *The wait here is way too long. I find a lot of difficulty in going to the doctors here. . . They put you in a room and they go and sleep or I don't know what. Maybe they go to eat. (1B)* |
| | • *We didn't call the ambulance. When his dad took him, he waited and waited and waited and waited, but it never got to his turn that time. He came back home. (3B)* |
| | • *If they explained all that to me, it would help a lot! I am sitting waiting, and I do not know why I am waiting (9B)* |
| | • *Since I am forced to come, I shouldn't be made to wait 5 or 6 hours just to call my son in. I didn't like that at all. I went for the first time and hopefully for the last time. I hope I never go again. I didn't like the experience. This thing is called EMERGENCY. (6B)* |
| Lack of Diagnostic Testing and Prescriptions | • *They don't prescribe it. No matter how much I asked them for it, they don't prescribe it. (3B)* |
| | • *I waited for a long time that day and at the end of it all, they didn't even give me any medicine. (1B)* |
| | • *They just put us there and left. All morning they had us waiting outside. When they finally called us in they still left us waiting in a room. Did they not want to give my daughter anything? Did they not want to do any imaging? An ultrasound? (4B)* |
| | • *[Back home], if we have an emergency we go immediately to the emergency room and they take care of you right away. They give you the medicine suitable for your ailment. . .They won't give you a pain relievers just to ease the symptoms. Here [in America] they just give you pain relievers all the time. [4A]* |

### Emergency department wait times

Syrian refugee parents shared frustration with the length of time waiting in emergent clinical settings, before and after being triaged. Some reported that the lengthy wait time hindered their access to medical care, as they often left without being seen due to constraints of employment and child care. Refugees detailed a desire to understand the reasons behind lengthy wait times, saying, "If they explained all that to me, it would help a lot! I am sitting waiting, and I do not know why I am waiting." Refugees did not comment that they perceived any bias directed towards them regarding wait times.

### Lack of diagnostic testing and prescriptions

Syrian refugee parents felt dissatisfied with the lack of diagnostic testing, both labs and imaging. They reported that in their home country, they more often received prescriptions for their children's ailments. They reported that their requests for prescriptions were not satisfied even after lengthy waiting, "At the end of it all, they didn't even give me any medicine."

### Theme 4: Resilience behaviors address perceived health barriers

Through the aforementioned perceived health barriers and challenges, parents exemplified resilience with the following facilitators to coping, as seen in Table 4.

### Faith

Faith in God was regarded as one method of managing stressors. Parents emphasized that through faith in God's plan and reading the Quran, their families experienced some level of

**Table 4. Parents use resilience behaviors to overcome barriers.**

| | |
|---|---|
| **Faith** | • *Reading the Quran is healing as well. It will put one at ease. That is what we do to feel better. . . God is the one that sends down the illness and He is the one that can cure it. (5B)* |
| | • *We have no one to turn to other than God. (4A)* |
| | • *Thank God we pray and we call out to God, and we are trying to be patient. Hopefully God will make it easy. There is nothing more we can do than that. Thank God. This is not something small, faith. (9B)* |
| | • *We are relying on God and doing what we can. (3A)* |
| | • *We go to the mosque. We pray together. We meet one another. (5A)* |
| **Seeking Knowledge** | • *We have to find a way to learn. (8A)* |
| | • *I want to know more about my son's disease, and I want to know what measures are taken to keep him healthy. (7A)* |
| | • *If a person sets their mind to it, they will learn. I am trying. (2B)* |
| **Community Resources** | • *Those two friends are the most ones that help. They know English and I turn to them a lot. I depend on them. Especially if there is a paper that I can't read. I send it to them. Everything. (1B)* |
| | • *If we had any questions, the Arab community answers it for us. They help even more [than the resettlement agency]. (2B)* |
| | • *One time there was a man from the Arab community. He came and took us, me and my kids. One time they also sent me someone from the community. A lady came and took me to the hospital. (7B)* |
| | • *And [the Syrian community] helped us to get to know the streets because, you know, it's a new city. . .They helped me get a driver license. They also, may God reward them, got us a car. And all of my matters became good. I started to go and come. [7A]* |
| **Home Remedies** | • *We first turn to herbs for remedies. (2A)* |
| | • *In our country, we can even use herbs as medicine. We use this or that. We treat them with herbs. Here they don't. They don't know. (4A)* |
| | • *If I find that my child is vomiting a lot, I would immediately treat him myself. . .We have some herbs. We boil them and give the child some to drink. (1A)* |

healing. One mother stated, "*Reading the Quran is healing as well. It will put one at ease. That is what we do to feel better. . . God is the one that sends down the illness and He is the one that can cure it.*" In some instances, participants remarked that there was no one to aid them besides God, and that their faith is strong but also requires patience. Parents expressed a balance between reliance on God and learning to survive, and that God commands them to seek knowledge.

### Knowledge seeking

A theme of acquiring knowledge and a desire to learn was evident in all participants' responses. They expressed a great need to learn in order to survive. "If a person sets their mind to it, they will learn. I am trying," one mother stated. Parents articulated that they did not want to remain uninformed about their children's diseases and illnesses. They desired for doctors to not only explain the disease but also its alleviating and exacerbating factors.

### Community resources

Parents asserted that the Arab and Muslim community help one other, especially upon new refugees arrival. Participants relied on the local Arab community for answers, transportation, translating assistance, and general emotional support. One participant remarked that the Arab community has been more supportive than the resources they received from the refugee settlement agency or the doctors and nurses at local hospitals. Participants welcomed support from volunteer agencies with specific tasks such as employment, basic needs, or ESL courses. Female participants consistently cited social support systems created within the community as a source of empowerment to manage their family's health needs.

### Home remedies

The remedies parents used in their country of origin were employed to attempt to cure children's illness prior to doctor visits. Some refugees stated that herbs can function as medicine and can cure an array of ailments, reporting "*We first turn to herbs for remedies.*" For example, rosemary was boiled for stomachaches, and garlic was utilized to decrease blood pressure. However, in America, refugees believe that doctors are unaware of the perceived benefit of using herbs as a medicine, saying "*We treat them with herbs. Here they don't. They don't know.*'

## Discussion

This qualitative study explored the perspectives, beliefs, and practices of Syrian refugee parents seeking healthcare for their children in a non-traditional migration city. For them, resilience behaviors tempered the systemic and individual health barriers.

### Stressors and health barriers preclude parental health seeking behaviors

Syrian refugee parents' psychological stressors negatively affect their health seeking behaviors for their children, making them less likely to seek care for their children's health. Psychological stressors were discussed equally among male and female participants. These findings underscore the importance of providers' awareness of the parental stressors that affect their ability to manage their children's health and seek help for their children's illnesses [9, 13, 31]. Beyond the individual provider, systems need to recognize, identify, and act upon these stressors during care encounters. Perhaps a brief, screening social work referral and mental health screening should be considered for refugees seeking healthcare. This could help address potential stressors, such as housing, food insecurity, employment, and mental health [32]. The social

determinants of health may affect poor health outcomes and inappropriate healthcare utilization for Syrian refugee children [33]. When stressors are identified, care coordinators could advocate for the needs of the pediatric patients and their parents. Other systems-based considerations to bridge existing gaps include closer relationships between health care institutions and refugee resettlement agencies, consistent with current dialogue [34]. This study begins to more fully characterize these issues, but further research is needed to develop community-engaged interventions to address parental stressors, implement, and evaluate such interventions.

## Health barriers hinder parental health seeking behavior

Syrian refugee parents also reported specific systemic barriers to managing their children's health. This suggests that the way refugee families are currently oriented to our health system may be faulty [9, 35, 36]. Participants' description of inadequate transportation and translation services depict fundamental gaps in quality healthcare. Complicating this, families have a lack of health literacy, both in terms of how to make an appointment and where to seek care (emergency department, urgent care, or primary care provider). These findings underscore the many foundational opportunities to improve care and parental perception of the barriers at hand. Studies on care coordination with complex care children, as well as with Bhutanese refugees, demonstrate its efficacy [9]. Care coordination may be more effective with community health workers, who embody the cultural competence to bridge gaps between providers and refugee parents, serving as cultural brokers [9, 37]. The establishment of a medical home for this population is a key step to coordinating care. In a city of non-traditional migration, a medical home specific to refugees is perhaps an important step to ensure quality care.

Furthermore, evidence suggests that in-person translation services for refugees are superior to phone or video translation [37]. With these findings in mind, a focus on care coordination, health literacy, and improved translation services may be important building blocks in creating more effective programs that address health barriers perceived by refugee families.

## Dissatisfaction with the U.S. healthcare system is prevalent

Parental dissatisfaction with the healthcare system is consistent with prior research in the general public—longer wait times in the emergent setting relates to dissatisfied patients and families [38–40]. Likewise, poor transition of information in the emergent settings leads to increased dissatisfaction and feelings of poor health management [40, 41]. Interestingly, Syrian refugees desire increased testing and treatments, aligning with previous findings that a physician who cares orders more testing and prescribes medications [42]. Medical providers should consider parental expectations and counsel refugee them on reasons behind pediatric lack of testing and treatment.

## Resilience behaviors address perceived health barriers

While the health barriers participants described in this study suggest a significant deviation from quality healthcare, the Syrian refugees in this study also described coping skills that helped them navigate these perceived barriers–resilience behaviors.

Seeking help from a network of community support is common among other refugee populations, for whom communal living may be the norm [9, 36, 43]. Healthcare providers and resettlement agencies could foster partnerships with social services and faith-based community agencies to provide Syrian refugees with the support they need to thrive in their new homes. This partnership begins with an awareness among health care providers that these parents have surpassed significant life challenges, and in doing so, they have gained a skillset to survive.

Systemic incorporation of community health workers in refugee resettlement could encourage this parental resilience and simultaneously bridge cultural gaps [9, 44]. Ultimately, community-academic partnerships deriving interventions could allow for improved understanding of how one's own community can be utilized to improve the health of Syrian refugee children [45].

Health care providers can consider the various ways Syrian refugees heal. Incorporation of home remedies and faith-based interventions, such as prayer and herbs, into care plans may potentially improve adherence and parental satisfaction [46]. Pediatricians should recognize and acknowledging that faith is an important facilitator to parental stressors.

It is known that "enhancing an individual's resilience" in the resettlement process is "more cost effective than long-term provision of social support" [32]. Likewise, "resilience oriented treatment" successfully addresses mental health of traumatized refugees [47]. Applying these concepts by building on the parental resilience behaviors may improve the health of pediatric refugees.

## Limitations

Like other qualitative research methods, interviews are limited by their non-random selection. Parental dyads were interviewed, which otherwise may have limited results. However, the captured breadth of experiences of 90% of Syrians resettled to Cincinnati allowed for credibility. This study's findings are limited to Syrian refugee parents, living in the Greater Cincinnati area within the first few years of arrival. Because of the recent timing of the Syrian Civil War and the timing it takes for resettlement, the earliest refugees were resettled two years prior to the study's initiation; therefore, it is not possible to identify participants who have been present for a longer time period. There was no non-inclusion, as all Syrian Refugees with children in the city were interviewed. Additionally, we recognize that it may be too soon for the respondents to understand the way the health system works; however, this is part of the reason why this study is important, because it identifies the barriers for those who have recently settled into the country. Our findings are not intended to be generalizable. Face to face in-depth interviews were conducted by two Syrian-Americans which allowed participants a safe, comfortable space to express themselves in Arabic, however during the translation and transcription process, words may have been altered.

## Conclusion

Pediatric refugees, as with most pediatric patients, likely thrive best in a primary care setting that is "accessible, comprehensive," and culturally competent [44, 48, 49]. Achieving this goal for refugee patients may look different than that for non-refugee patients [9, 50], and cultural competence may need to be couture for each specific population. Psychosocial and environmental stressors must be addressed to improve parental health seeking behaviors. Culturally relevant care that elicits parental expectations, capitalizes on resilience behaviors, and targets perceived barriers is likely to improve parental satisfaction and pediatric refugee health. Consistent with current best practices, establishment of a medical home through community partnerships and systemic care coordination may be the solution [45].

## Supporting information

**S1 File. Semi-structured interview guide.**
(DOCX)

**S2 File. Statement of positionality.** Statement of positionality of the authors.
(DOCX)

**S3 File. Full dataset, compiled.**
(DOCX)

# Acknowledgments

Special thanks to Rahma Social Services for their assistance with recruitment.

# Author Contributions

**Conceptualization:** Riham M. Alwan, Ahmed Beydoun, Tasnim Salem, Lisa M. Vaughn.

**Data curation:** Riham M. Alwan, Sevsem Cicek-Okay, Ahmed Beydoun, Tasnim Salem.

**Formal analysis:** Riham M. Alwan, Sevsem Cicek-Okay, Sarah Jernigan, Ahmed Beydoun, Tasnim Salem, Lisa M. Vaughn.

**Funding acquisition:** Riham M. Alwan.

**Investigation:** Riham M. Alwan, Ahmed Beydoun.

**Methodology:** Riham M. Alwan, Daniel J. Schumacher, Sevsem Cicek-Okay, Sarah Jernigan, Tasnim Salem, Lisa M. Vaughn.

**Project administration:** Daniel J. Schumacher, Ahmed Beydoun.

**Resources:** Lisa M. Vaughn.

**Supervision:** Daniel J. Schumacher, Lisa M. Vaughn.

**Validation:** Riham M. Alwan, Sarah Jernigan, Lisa M. Vaughn.

**Visualization:** Riham M. Alwan, Sevsem Cicek-Okay, Tasnim Salem.

**Writing – original draft:** Riham M. Alwan, Sevsem Cicek-Okay, Sarah Jernigan.

**Writing – review & editing:** Riham M. Alwan, Daniel J. Schumacher, Sevsem Cicek-Okay, Sarah Jernigan, Ahmed Beydoun, Tasnim Salem, Lisa M. Vaughn.

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
