## [Decision Letter · Decision Letter 0]

3 Jan 2020

PONE-D-19-31988

Beliefs, perceptions, and behaviors impacting healthcare utilization of Syrian refugee children

PLOS ONE

Dear Dr Alwan,

Thank you for submitting your manuscript to PLOS ONE. After careful consideration, we feel that it has merit but does not fully meet PLOS ONE’s publication criteria as it currently stands. Therefore, we invite you to submit a revised version of the manuscript that addresses the points raised during the review process.

We would appreciate receiving your revised manuscript by Feb 17 2020 11:59PM. To enhance the reproducibility of your results, we recommend that if applicable you deposit your laboratory protocols in protocols.io, where a protocol can be assigned its own identifier (DOI) such that it can be cited independently in the future. For instructions see: http://journals.plos.org/plosone/s/submission-guidelines#loc-laboratory-protocols

We look forward to receiving your revised manuscript.

Kind regards,

Vijayaprasad Gopichandran

Academic Editor

PLOS ONE

Journal Requirements:

3. Please include your tables as part of your main manuscript and remove the individual files. Please note that supplementary tables (should remain/ be uploaded) as separate "supporting information" files

Reviewers' comments:

Reviewer's Responses to Questions

**Comments to the Author**

1. Is the manuscript technically sound, and do the data support the conclusions?

Reviewer #1: Partly

Reviewer #2: Yes

2. Has the statistical analysis been performed appropriately and rigorously? 

Reviewer #1: N/A

Reviewer #2: N/A

3. Have the authors made all data underlying the findings in their manuscript fully available?

Reviewer #1: Yes

Reviewer #2: Yes

4. Is the manuscript presented in an intelligible fashion and written in standard English?

Reviewer #1: Yes

Reviewer #2: Yes

5. Review Comments to the Author

Reviewer #1: 1. Introduction:

a) Line 96-98 should clearly mention that the authors are looking at the impact of beliefs, perspectives and practices of parents in seeking care for their children.

2. Methods:

a) It would be useful to elaborate more on the immigrant support infrastructure that exist in other US cities and contrast it with Cincinnati. This would give more idea of the context. Lines 106-107 mention the lack of infrastructure in non traditional city of migration but not much information on these.

b) Line 107-108 on poorer health outcomes should come under the introduction section.

c) What is the eligibility criteria for inclusion, please elaborate. Was relocation in the past two years the only criteria for inclusion? Only the primary care givers were interviewed or other elderly people in the household?

3. Data Analysis:

a) Line 132-135: Please refer to standard books on qualitative analysis and try to explain using technical terms here.

b) Line 132: Elaborate on the process of transcribing. Were they first translated and then transcribed or was it done simultaneously?

c) Reading the analysis section gives an impression that Theoretical sampling was used. Can you please mention if that was the case or coding was done after all the interviews were completed.

d) It would be useful to state the positionally of the Researcher as it would help the readers to interpret the findings

4. Results:

a) It would be better if the quotes in different tables are incorporated in the text so that it is easy for the reader to relate to what the authors are trying to convey. Can have one table mentioning only the themes and subthemes.

b) There is overuse of the word “Parents” in describing the findings. The sentences can be reframed.

c) Reasons for missed appointments. Need more explanation rather than just mentioning transport, language and navigation. It would be helpful to group the barriers into one that is at the level of parents, community and health system.

d) Specific difficulties in navigating public transport system- Explain the difficulties faced as explained by the respondents

e) Can highlight about the interpreter system that is existing in clinical set up like who are these interpreters? Are they from similar regions?

f) Can you cal it lack of health literacy or the inability of the system to explain things to them. In table two the authors talk about respondents not been given any explanation for the medications. This is a fault of the system and not of the respondents/parents. Wrt sick children is it because the parents dont know or the system is difficult to navigate. Reading from table 2 indicates that they dont know whether to take the child to the hospital or just communicate with the provider. Not knowing which medical centre to take the child to may not be a fault of the parents. They have been there for less than 2 years and may be its not their fault that they have not been aware or informed of the health care facilities that exist there.

g) Emergency waiting time: Is there any explanation on why they were made to wait for long? Is there any kind of discrimination as perceived by them?

h) Faith: resilience for health care barriers or generally for the refugee status? Does Faith influence the pattern of care seeking? The section on faith seems like a misfit under theme 4

i) Home remedies: The doctors in America were unaware of the different home remedies but do they dismiss them or discourage them from doing it? When do the parents use home remedy? Is it before visiting the doctor or taken along with allopathic medicines.

j) While reading the text it does seem like the access related issues are both at the level of the individuals as well as the health system. It would be useful to use an existing framework for access to organise the findings. One possible framework that the authors could look at is that of Levesque (Levesque J-F, Harris MF, Russell G. Patient-centred access to health care: conceptualising access at the interface of health systems and populations. International Journal Equity Health. 2013;12:18). This is not the only framework but there are many and the authors could refer to them and structure the results based on that.

k) The results section seem to be disconnected from each other. There needs to be a connection

l) Is there any information in the interviews to suggest that the experience of the parents with the health system affect the care seeking for children for example any experience of discrimination or personal experiences

m) The differences in the experience with the health care system in the place of origin and in the current place has not been reflected in the paper

5. Discussion

a) The discussion like the result section is disconnected from each other. There should be a continuity

b) Faith component is absent in the discussion section and not explored much

6. Limitations

a) The major limitation is the inclusion criteria i.e relocation in the past two years. It may be too early for the respondents to understand the way the health system works in the current setting. This needs to be stated.

7. Conclusions

a) When you talk about care that is culturally competent you need to mention about the cultural competence in general to other subpopulations and if that exists in the current system. If it is non existent then one cannot expect the system to be more receptive to refugees and their needs and this is a major problem.

8. General Comments

a) Please verify the format for in-text citations. The number in parenthesis should be placed before the full stop. Kindly look into Plos one guidelines.

b) The Interview guide appears to be more rigid and structured like a quantitative survey. It is good to organise into sections but the questions need to be free flowing.

Reviewer #2: 1. What is the methodological orientation of the study?

2. I find the some sentences that reflect quantitative sensibilities. These may be rewritten to fit the qualitative paradigm better. For instance –

a. Page 12 – line 148 – “Ninety percent of eligible Syrian refugee families were enrolled.” – a sentence that seems to reflects “coverage” from a “denominator”. I suggest reporting on how saturation was achieved and how potential redundancy of further data collection was confirmed.

b. Page 13 lines 161-162 -“Findings did not differ among educational level, employment status, or Syrian area of origin.” The write up needs to inform better on the diversity of codes and themes, or lack of the same.

3. Discuss how the research team approached reflexivity - particularly the interviewers. Interviewer AS - I am not able to discern who among the authors this indicates. (Page 11 Line 123)

4. Did the transcription follow the translation? If so, how was translation done? (Page 11, line 132)

5. Since the purpose is not quantitative estimation, sample size is not a limitation of the study. Limitations specific to the qualitative paradigm need to be reported – e.g. influence of the researcher’s gaze on the interpretations, non-inclusion of people of a certain category/ circumstance, whether saturation was possibly a result of the methodology etc. (Page 22, Lines 372-373)

6. “Garlic was utilized to decreased blood pressure.” (Page 18, 292-293). Did you mean “to decrease”? How did blood pressure come into a study on healthcare utilization of children? Was it one of the health problems diagnosed or was it a perceived health issue? Or, did issues related to health care of adults find its way into the discourse? I suggest retaining this information, but with some explanation on it.

7. A medical home is given as the conclusion that follows from the study (Page 23, lines 389-390). While medical home is mentioned in the introductory part, concluding in this manner without dwelling on it in the discussion seems inadequate.

6. PLOS authors have the option to publish the peer review history of their article (what does this mean?). If published, this will include your full peer review and any attached files.

Reviewer #1: Yes: Bevin Vijayan

Reviewer #2: Yes: Ravi Prasad Varma

---

## [Author Response · Author response to Decision Letter 0]

16 Jul 2020

PONE-D-19-31988

Beliefs, perceptions, and behaviors impacting healthcare utilization of Syrian refugee children

PLOS ONE

Thank you for your kind reminder. We have performed formatting edits and now the manuscript should meet PLOS ONE’s style requirements.

Thank you for your kind reminder. We have uploaded the dataset.

3. Please include your tables as part of your main manuscript and remove the individual files. Please note that supplementary tables (should remain/ be uploaded) as separate "supporting information" files

Thank you for your kind reminder. We have made the requisite changes.

Thank you for your kind reminder. We have made the requisite changes.

Reviewer Comments to the Author

Reviewer #1: 

1. Introduction:

a) Line 96-98 should clearly mention that the authors are looking at the impact of beliefs, perspectives and practices of parents in seeking care for their children.

Thank you for your recommendation. We have clarified the sentence. Lines 55-57

2. Methods:

a) It would be useful to elaborate more on the immigrant support infrastructure that exist in other US cities and contrast it with Cincinnati. This would give more idea of the context. Lines 106-107 mention the lack of infrastructure in non traditional city of migration but not much information on these.

Thank you for your recommendation. We have highlighted some infrastructure that lacking in non-traditional cities of migration. Lines 65-70

b) Line 107-108 on poorer health outcomes should come under the introduction section.

Thank you for your recommendation. We have included it in the introduction section as well. Lines 54-55

c) What is the eligibility criteria for inclusion, please elaborate. Was relocation in the past two years the only criteria for inclusion? Only the primary care givers were interviewed or other elderly people in the household?

Our inclusion criteria were: being of Syrian origin; having dwelled in Syria prior to the civil war; having cohabitating children or grandchildren below the age of 18; refugee status; Medicaid enrolled or eligible; and migration within the last two years. Participants who did not speak Arabic or English were excluded. Only primary care givers were interviewed in the study, as no other elderly house-hold members were present.

We have updated the manuscript, Lines 77-82

It is important to note that the refugee who had been settled in the area for the longest time was relocated two years prior to the initiation of the study, which was part of the wisdom in having the inclusion criteria being resettlement less than or equal to two years. Lines 394-397

3. Data Analysis:

a) Line 132-135: Please refer to standard books on qualitative analysis and try to explain using technical terms here.

Thank you for your recommendation. We have updated this section to more technically describe the iterative process of inductive thematic analysis. Lines 99-116

 b) Line 132: Elaborate on the process of transcribing. Were they first translated and then transcribed or was it done simultaneously?

The Arabic conversation was audio recorded, then simultaneously translated and transcribed into English. Lines 99-100

c) Reading the analysis section gives an impression that Theoretical sampling was used. Can you please mention if that was the case or coding was done after all the interviews were completed.

Coding was done subsequently. See lines 103-107

d) It would be useful to state the positionally of the Researcher as it would help the readers to interpret the findings 

Thank you for your recommendation. We have provided a statement of positionality, with emphasis on the senior author who coordinated the study. Lines 93-94, Supplemental File 1

4. Results:

a) It would be better if the quotes in different tables are incorporated in the text so that it is easy for the reader to relate to what the authors are trying to convey. Can have one table mentioning only the themes and subthemes.

Thank you for the recommendation. In the spirit of transparency and the purist qualitative methodology, we have included more direct quotes into the manuscript. 

b) There is overuse of the word “Parents” in describing the findings. The sentences can be reframed.

Thank you for the recommendation. We have performed revisions and feel that the manuscript reads more smoothly now.

c) Reasons for missed appointments. Need more explanation rather than just mentioning transport, language and navigation. It would be helpful to group the barriers into one that is at the level of parents, community and health system.

We are cautious of oversimplification by stratifying into ‘parents,’ ‘community,’ and ‘health.’ For example, difficulty with transportation is confounded by problems at an individual health level (lack of a vehicle, lack of money for gas, lack of knowing how to drive), at a community level (restricted public transportation), and a systematic level (no Ubers). It is common for a parent to miss their appointments because they have no car for transportation (financial), cannot get a license because they have no comprehension of the laws and language, no assistance to help them through the process, no public transportation available to their household, and no community support to help them transport, and the health system does not aid with transportation – in this particular city. In other cities in America, specific hospital systems have learned how to transport patients to/from appointments. 

Lines 191-201

d) Specific difficulties in navigating public transport system- Explain the difficulties faced as explained by the respondents

Thank you for your suggestion. We have done the requisite changes. Lines 204-212

e) Can highlight about the interpreter system that is existing in clinical set up like who are these interpreters? Are they from similar regions?

Due to the diversity of the colloquial Arabic and limited availability, it is clear that the interpreter services did not provide the adequate set-up for the participants to feel both understood as well as to understand. The interpreter services are provided by the hospital system, outsourced from a company, and so we are unable to comment on the who the interpreters are nor their region of origin. Lines 215-219

f) Can you call it lack of health literacy or the inability of the system to explain things to them. In table two the authors talk about respondents not been given any explanation for the medications. This is a fault of the system and not of the respondents/parents. Wrt sick children is it because the parents dont know or the system is difficult to navigate. Reading from table 2 indicates that they dont know whether to take the child to the hospital or just communicate with the provider. Not knowing which medical centre to take the child to may not be a fault of the parents. They have been there for less than 2 years and may be its not their fault that they have not been aware or informed of the health care facilities that exist there.

Thank you for your insight, and we are in agreement.

The refugees lack of knowledge is not their own fault, but due to systematic barriers that we are elucidating in this manuscript. It is important to note that health literacy is defined by the ability of the person to find information and services, communicate their needs, understand what is being communicated to them and the choices available, and then make a decision based on this information. We have defined health literacy overtly for the readers, and make notes that it is not associated with education. Lines 222-229

g) Emergency waiting time: Is there any explanation on why they were made to wait for long? Is there any kind of discrimination as perceived by them?

Our participants did not perceive any bias towards them in regards to their wait times. They are describing one of the barriers to access for care within the American healthcare system. We have updated the manuscript: Lines 254-255

h) Faith: resilience for health care barriers or generally for the refugee status? Does Faith influence the pattern of care seeking? The section on faith seems like a misfit under theme 4

Thank you for your question.

Faith was a facilitator for coping with challenges in both healthcare as well as their refugee status. Our emphasis was on health barriers and not their refugee status, and so we did not delve into coping with their refugee status.

We did not see evidence that faith influenced the pattern for care seeking, as in patients with faith did not forgo seeking medical care. Instead it was used as a mechanism for acceptance for their situation, and thus it is a mechanism for resilience.

This is additionally why we disagree that it is a misfit for Theme 4: Resilience, as it is used as a coping mechanism which helps build resistance; the parents described using faith to enhance their patience and their ability to cope with illness. Additionally, resilient parents are more likely to seek care and thrive in challenging medical situations.

i) Home remedies: The doctors in America were unaware of the different home remedies but do they dismiss them or discourage them from doing it? When do the parents use home remedy? Is it before visiting the doctor or taken along with allopathic medicines.

The participants did not mention that they were discouraged from using their home remedies. The parents used the home remedies prior to the doctor’s visit as well as along with the allopathic medications. Lines 300-301

j) While reading the text it does seem like the access related issues are both at the level of the individuals as well as the health system. It would be useful to use an existing framework for access to organise the findings. One possible framework that the authors could look at is that of Levesque (Levesque J-F, Harris MF, Russell G. Patient-centred access to health care: conceptualising access at the interface of health systems and populations. International Journal Equity Health. 2013;12:18). This is not the only framework but there are many and the authors could refer to them and structure the results based on that. k) The results section seem to be disconnected from each other. There needs to be a connection

Thank you for your recommendation. We agree that barriers seem to follow a pattern of individual, systematic, societal, etc. However, this study was designed as a critical ethnography, and did not use grounded theory methodology. Therefore, we cannot develop a theoretical framework. Data was analyzed via inductive thematic analysis, and our results are organized based on the themes that emerged during data analysis.

l) Is there any information in the interviews to suggest that the experience of the parents with the health system affect the care seeking for children for example any experience of discrimination or personal experiences

It is true that the parent’s experience with the health system will influence how they seek care for their children. However, the scope of this manuscript was to focus on the health seeking behavior for their children, and did not discuss the parent’s own experiences.

m) The differences in the experience with the health care system in the place of origin and in the current place has not been reflected in the paper

Thank you for your recommendation. We are currently working on a separate manuscript that discusses parent’s past experiences with current experiences. 

5. Discussion

a) The discussion like the result section is disconnected from each other. There should be a continuity

Thank you for your recommendation. Our discussion is organized based on the different themes that were identified in the course of our research. If the themes are disconnected then they are individual themes to be understood separately.

b) Faith component is absent in the discussion section and not explored much

Thank you for your recommendation. Within the context of a secular society, we are limiting our recommendations to faith-based partnerships, cognizance of the importance of faith within this community, and coping interventions for this unique population. Lines 367-369, 376-379

6. Limitations

a) The major limitation is the inclusion criteria i.e relocation in the past two years. It may be too early for the respondents to understand the way the health system works in the current setting. This needs to be stated.

Thank you for your observation.

Because of the recent timing of the Syrian Civil War and the timing it takes for resettlement, the earliest refugees were resettled two years prior to the study’s initiation; therefore it is not possible to identify a longer time period. We have updated the manuscript accordingly. 

Additionally, we recognize that it may be too soon for the respondents to understand the way the health system works; however, this is part of the reason why this study is important, because it identifies the barriers for those who have recently settled into the country. 

Lines 391-397

7. Conclusions

a) When you talk about care that is culturally competent you need to mention about the cultural competence in general to other subpopulations and if that exists in the current system. If it is non existent then one cannot expect the system to be more receptive to refugees and their needs and this is a major problem.

Thank you for question.

Indeed, there is a culture of general cultural competence at this institute. Being a top ranked children’s hospital in the USA allows it to attract many families from international communities who seek specialized care. The center which coordinates their arrival is called ‘Destination Excellence,’ and provides a 24/7 helpline, assistance with housing, assistance with schooling, and other basic needs for international families. Unfortunately, refugee families who are already resettled in the USA do not have access to Destination Excellence. This is part of the reason that emphasizes the importance of this paper. 

https://www.cincinnatichildrens.org/patients/visit/international

To this end, we mention how cultural competence may not have to be couture for each specific population. Line 340-342, 406-407

8. General Comments

a) Please verify the format for in-text citations. The number in parenthesis should be placed before the full stop. Kindly look into Plos one guidelines.

Thank you for the reminder. We have corrected this error

b) The Interview guide appears to be more rigid and structured like a quantitative survey. It is good to organise into sections but the questions need to be free flowing.

The interview guide is semi-structured intentionally. These were the general questions that were used to guide the conversations, which was free-flowing in the participants native language. 

Reviewer #2:

1. What is the methodological orientation of the study?

Thank you for your question. We have included a statement of positionality which addresses the methodological orientation of the study, with emphasis on the senior author who coordinated the study Lines 93-94, Supplemental File 1

2. I find the some sentences that reflect quantitative sensibilities. These may be rewritten to fit the qualitative paradigm better. For instance –

a. Page 12 – line 148 – “Ninety percent of eligible Syrian refugee families were enrolled.” – a sentence that seems to reflects “coverage” from a “denominator”. I suggest reporting on how saturation was achieved and how potential redundancy of further data collection was confirmed. 

Thank you for your recommendation. We agree that for a purist qualitative study should not be using quantitative nomenclature. However, our target audience are physicians who have a quantitative background, and so it may serve as an important point of comprehension without subtracting from the merits of our study.

As a standard iterative process for qualitative research, we were monitoring for saturation. Saturation was achieved when we identified repetitive themes without any novel information. For example, every parent discussed poor housing, challenges with interpreter services, and their own parental anxieties. A lengthy process of member checking was used to enhance credibility and trustworthiness of the data.

The manuscript has been updated to reflect these thoughts. Lines 99-116

b. Page 13 lines 161-162 -“Findings did not differ among educational level, employment status, or Syrian area of origin.” The write up needs to inform better on the diversity of codes and themes, or lack of the same.

Unfortunately, we disagree on the necessity of presenting inter-coder reliability data, as the three coders consistently met to review the codes, categories, and themes. The code book was an iterative process with evolved with input from all three coders simultaneously, and so inter-coder reliability may not be calculatable. Additionally, including an inter-coder reliability contradicts the interpretive agenda of qualitative research.

3. Discuss how the research team approached reflexivity - particularly the interviewers. Interviewer AS - I am not able to discern who among the authors this indicates. (Page 11 Line 123)

Thank you for your question. We have included a statement of positionality, which touches upon the team’s reflexivity.

Thank you for your observation. Interviewer AS is a typo – it should be AB, which corresponds to our 5th author. 

4. Did the transcription follow the translation? If so, how was translation done? (Page 11, line 132)

Interviews were audio-recorded in Arabic then simultaneously translated and transcribed into English. Two bilingual researchers (RA, AB) double checked the transcriptions for accuracy and content. Lines 99-105

5. Since the purpose is not quantitative estimation, sample size is not a limitation of the study. Limitations specific to the qualitative paradigm need to be reported – e.g. influence of the researcher’s gaze on the interpretations, non-inclusion of people of a certain category/ circumstance, whether saturation was possibly a result of the methodology etc. (Page 22, Lines 372-373)

Thank you. We did not intend to say that sample size was a limitation, and have adjusted our verbiage.

The researcher’s gaze on interpretations is discussed in our now included positionality statement. 

There was no non-inclusion, as all Syrian Refugees with children in the city were interviewed. Lines 394-395

6. “Garlic was utilized to decreased blood pressure.” (Page 18, 292-293). Did you mean “to decrease”? How did blood pressure come into a study on healthcare utilization of children? Was it one of the health problems diagnosed or was it a perceived health issue? Or, did issues related to health care of adults find its way into the discourse? I suggest retaining this information, but with some explanation on it.

Thank you for identifying the typo – decreased is supposed to be decrease.

In regards to the blood pressure, one of the children did have renal disease and high blood pressure, but in general the adults were using an example of home remedies for their own health and translating it to how they use home remedies to children.

7. A medical home is given as the conclusion that follows from the study (Page 23, lines 389-390). While medical home is mentioned in the introductory part, concluding in this manner without dwelling on it in the discussion seems inadequate.

 Thank you for your recommendation. We have updated the discussion. Lines 342-344, 410-412

---

## [Editor Report · Decision Letter 1]

21 Jul 2020

Beliefs, perceptions, and behaviors impacting healthcare utilization of Syrian refugee children

PONE-D-19-31988R1

Dear Dr. Alwan,

We’re pleased to inform you that your manuscript has been judged scientifically suitable for publication and will be formally accepted for publication once it meets all outstanding technical requirements.

Kind regards,

Vijayaprasad Gopichandran

Academic Editor

PLOS ONE
---

## [Editor Report · Acceptance letter]

30 Jul 2020

PONE-D-19-31988R1 

Beliefs, perceptions, and behaviors impacting healthcare utilization of Syrian refugee children 

Dear Dr. Alwan:

I'm pleased to inform you that your manuscript has been deemed suitable for publication in PLOS ONE. Congratulations! Your manuscript is now with our production department. 

Kind regards, 

on behalf of

Dr. Vijayaprasad Gopichandran 

Academic Editor

PLOS ONE